# Association between Urinary Phthalate Metabolites and Markers of Endothelial Dysfunction in Adolescents and Young Adults

**DOI:** 10.3390/toxics9020033

**Published:** 2021-02-06

**Authors:** Po-Ching Chu, Charlene Wu, Ta-Chen Su

**Affiliations:** 1Department of Environmental and Occupational Medicine, College of Medicine, National Taiwan University, No. 1 Ren’ai Road Section 1, Taipei 10051, Taiwan; pcchu@ntuh.gov.tw; 2Department of Environmental and Occupational Medicine, National Taiwan University Hospital, No. 7 Chung-Shan South Road, Taipei 10002, Taiwan; 3Global Health Program, College of Public Health, National Taiwan University, Taipei 10055, Taiwan; charlenewu@ntu.edu.tw; 4Division of Cardiology, Department of Internal Medicine, National Taiwan University Hospital, Taipei 10051, Taiwan; 5Institute of Environmental and Occupational Health Sciences, College of Public Health, National Taiwan University, Taipei 10055, Taiwan

**Keywords:** phthalates, di-n-butyl phthalate, microparticles, endothelial dysfunction, cardiotoxicity

## Abstract

Endothelial function is crucial in the pathogenesis of circulatory and cardiovascular toxicity; epidemiologic research investigating the association between phthalate exposure and endothelial dysfunction remains limited. We examined the associations between exposures to specific phthalates (di-2-ethylhexyl phthalate, DEHP; di-n-butyl phthalate, DnBP) and circulating endothelial and platelet microparticles (EMPs and PMPs) in adolescents and young adults. Of the 697 participants recruited, anthropometric measurements and health-related behaviors relevant to cardiovascular risks were collected and assessed. Urine and serum were collected and analyzed with liquid chromatography-tandem mass spectrometry (LC-MS/MS) and flow cytometry. Multiple linear regression indicated that increases in urinary concentrations of ΣDEHP and MnBP (mono-n-butyl phthalate), across quartiles, were positively associated with serum EMPs level (*p* for trend <0.001 and <0.001; β = 0.798 and 0.007; standard error = 0.189 and 0.001, respectively). Moreover, female and overweight subjects had higher MnBP, and males were more vulnerable to DnBP exposure compared to females. In conclusion, our results demonstrate a dose-response relationship between exposures to phthalates (ΣDEHP and MnBP) and microparticle formation (EMPs and PMPs) in adolescents and young adults. The findings indicate that exposures to phthalates of both low and high-molecular weight are positively associated with microparticle production, and might contribute to endothelial dysfunction; such damage might manifest in the form of atherosclerotic-related vascular diseases. Future in vivo and in vitro studies are warranted to elucidate whether a causal relationship exists between phthalate exposure and EMPs and PMPs.

## 1. Introduction

Phthalate exposure and toxicity is of major concern to public health; among these groups of environmental chemicals, di-2-ethylhexyl phthalate (DEHP) is a key component in consumer and personal care products worldwide. The production volume of dibutyl phthalate (DBP), another commonly used plasticizer, ranged between one and ten million pounds in the USA by 2020; in Asia, the demand for DBP remained high, with an estimated production of 7000 tons in Korea [1,2]. In the European Union (EU), production of DBP remained consistent at approximately 10,000 tons [3]. Therefore, in light of the abundant use of DEHP and DBP [1], the health hazards posed to the general population through inhalation, ingestion, and dermal contact with both natural and man-made products continue to exist. Based on the EU risk assessment report, DBP exposure poses general systemic toxicity to humans via repeated dermal exposure from aerosol forming activities [4].

Abundant animal study assessed the adverse health effects of phthalates, including endocrine disruption, reproductive toxicity, neurological, renal, and cardiotoxicity [5,6,7,8,9]. Chen et al. reported developmental toxicity from DBP and estrogenic endocrine disrupting activity from DEHP and DBP in zebrafish embryos. Furthermore, these phthalates could cause additional toxicity symptoms such as cardio edema [5]. Another animal toxicity study revealed that rats chronically exposed to high oral doses of DEHP exhibited many noncarcinogenic effects such as changes in clinical hematology and manifestations of increased liver lesions [7]. DEHP exposure during lactational period was also noted to impair insulin signal transduction and oxidation in the cardiac muscle of female progeny rats [8]. In utero exposure to DEHP caused long-term cardiovascular effects in male offspring rats [9]. A review indicated that there was a weak weight-of-evidence regarding the cardiovascular risk of phthalate exposure, and the major DEHP metabolite had cardiotoxic effects in humans [10]. Urinary DEHP level was positively associated with blood pressure in childhood, and the level of DEHP was associated with diabetes and serum insulin levels [11]. In additional to animal experiments, systemic review of both epidemiological and animal toxicity studies concluded that DEHP and DBP exposure may lead to adverse reproductive health outcomes in both males and females [6,10,12]. DEHP and DnBP had anti-androgenic effects in males [10,12], which could affect testicle function, and it may lead to male fertility disorders and testicular dysgenesis syndrome [12]. Additionally, the biochemical mechanism through which cardiotoxicity occurs have been reported. DEHP exposure in mice can induce oxidative stress, inhibit acetylcholinesterase activity, and lead to structural alternation of surrounding blood vessels, as reported in previous literature [13]; these disruptions provide evidence that exposure to DEHP may promote the progression of atherosclerosis. Furthermore, Ferguson and associates indicated that mono-n-butyl phthalate (MnBP), the metabolite of di-n-butyl phthalate (DnBP), and the metabolites of DEHP were significantly and positively associated with markers of oxidative stress (bilirubin) and inflammation (fibrinogen, alkaline phosphatase) based on data from participants in the National Health and Nutrition Examination Survey [14].

Endothelial and platelet microparticles (EMPs, CD31+/CD42a−; PMPs, CD31+/CD42a+) have been used as surrogate markers of endothelial dysfunction and cardiovascular diseases [15,16,17,18,19,20]. Previous literature identified that clinically, EMPs, which can be shed from impaired and/or apoptotic endothelial cells, are indicative of cardiovascular damage [15,21]. We noted few articles that explored the association between phthalate exposure and microparticle formation as an indicator of cardiotoxicity; Kataria and colleagues reported a negative association between the metabolites of high-molecular weight phthalate and EMPs [22]. Our previous study in adolescents and young adults found a positive association between mono-2-ethylhexyl phthalate (MEHP), a metabolite of DEHP, and endothelial and platelet microparticles [19]. However, the association between exposure to commonly used phthalates, such as ΣDEHP and DnBP, and microparticles is yet to be investigated, especially in adolescents and young adults. Thus, we hypothesize exposures to ΣDEHP and DnBP are likely to induce endothelial dysfunction, and initial damage may manifest in increased counts of circulating microparticles in serum; we propose a cross-sectional investigation in a cohort to elucidate whether exposures to DEHP and DnBP are associated with microparticle formation in adolescents and young adults.

## 2. Methods

### 2.1. Study Participants

The YOung TAiwanese Cohort (YOTA) was a nationwide screening program established from 1992 to 2000 for renal diseases among age 6–18 years in Taiwanese schools [23,24]. Participants engaging in the screening program were contacted for follow-up examination in 2006–2008. Among the participants who joined the screening program and categorized with elevated blood pressure, 303 residents of Taipei completed the follow-up examination. We randomly reached out to 6390 participants with normal blood pressure and recruited 486 participants from Taipei; the detailed methods are described in our previous study [24]. We further excluded 21 participants due to missing urine samples, and 71 subjects were eliminated because their urinary creatinine concentrations were below 0.3 g/L or above 3.0 g/L, which was based on guidelines published by the World Health Organization and National Institute of Health [25,26]. Therefore, 697 participants remained as our total study population. A diagram detailing the selection criteria of study subjects is presented in Figure 1. This study protocol was approved by the Research Ethics Committee of National Taiwan University Hospital. Informed written consent was obtained from each participant and their parents before they participated in the study.

### 2.2. Assessment of Personal and Anthropometric Data

Personal data were obtained from a structured and self-administered questionnaire, including basic demographic data (i.e., age, gender), lifestyle habits (i.e., cigarette smoking, alcohol consumption, physical activity), medication history, and household income. Smoking status was classified into two groups: (1) active smoker and (2) non-active smoker. Alcohol consumption was classified into two groups: (1) current alcohol consumption and (2) non-current alcohol consumption. Physical activity was classified into two groups: (1) regular activity and (2) non-regular activity. Body mass index (BMI) was evaluated as body weight divided by the square of body height (kg/m^2^); it was classified into two groups: (1) <24 and (2) ≥24. Household income was classified into two groups: (1) <50,000 New Taiwan dollar (NTD)/month, and (2) ≥50,000 NTD/month. Two sets of seated blood pressure were recorded after 5 min of rest with a mercury sphygmomanometer and appropriate cuff size. Hypertension was determined by receiving anti-hypertension medicines currently, or a mean blood pressure of more than or equal to 140/90 mm Hg.

### 2.3. Biochemical Measurements

Blood samples were obtained in the morning after participants had endured more than eight hours of fasting; subsequent serum samples were stored at −80 °C prior to analysis. The serum concentrations of glucose and total cholesterol were measured using an autoanalyzer (Technician RA 2000 Autoanalyzer, Bayer Diagnostic, Mishawaka, India). Diabetes mellitus was determined by the current use of oral hypoglycemic agents or insulin, or a fasting serum glucose of more than or equal to 126 mg/dL. Total cholesterol was classified into two groups: (1) <200 and (2) ≥200 mg/dL. The coefficient of variation for serum total cholesterol and glucose was under 3%.

### 2.4. Analysis of Urine Phthalate Metabolites

The first-void urine samples in the morning were collected from all participants to analyze urinary metabolites of phthalates, including MEHP, mono(ethyl-5-hydroxyhexyl) phthalate (MEHHP), mono(2-ethly-5-oxoheyl) phthalate (MEOHP), and MnBP. The urine samples were quantitatively analyzed by liquid chromatography-tandem mass spectrometric (LC-MS/MS); the detailed methods were the same as our previous study [27]. Internal quality control was performed applying pooled quality control urines and external quality assurance was assessed using the German External Quality Assessment Scheme for Biological Monitoring (G-EQUAS). The method detection limits of MnBP, MEHP, MEHHP, and MEOHP were 1.0, 0.7, 0.1, and 0.1 μg/L, respectively. Laboratory results below the limits of detection were recorded as the detection limit divided by the square root of two [28]. The detection rates for MEHP, MEHHP, MEOHP, and MnBP were 75.18%, 99.71%, 100%, and 99.71%, respectively. The urinary metabolite concentrations were adjusted with urinary creatinine.

### 2.5. Measurements of EMPs and PMPs

EMPs and PMPs were estimated with a flow cytometer using the method based on previous research [26]. The microparticles were measured simultaneously in citrated serum with a pair of fluorescent monoclonal antibodies: phycoerythrin-labeled anti-CD31 (BD Bioscience, San Jose, CA, USA ) and fluorescein isothiocyanate-labeled anti CD42a (BD Bioscience, San Jose, CA, USA). The values of the microparticles are reported as counts/μL.

### 2.6. Statistical Analysis

To determine total exposure to DEHP, we estimated urinary concentrations of the metabolites by summing the molar concentrations of MEHP, MEHHP, and MEOHP as described in previous research (hereafter referred to as ΣDEHP in μmol/g creatinine) [29,30]; all other phthalate metabolites analyzed in this study are expressed in μg/g creatinine [31,32,33,34,35,36,37,38,39]. The distribution of different phthalate metabolites according to basic characteristics were analyzed, with urine metabolites concentrations expressed as geometric means and 95% confidence intervals (95% CI). The characteristics included the following: age, gender, hypertension, diabetes mellitus, total cholesterol, body mass index, smoking status, alcohol consumption, physical activity, and household income. Similarly, the distribution of EMPs and PMPs among our study participants were analyzed. A Kruskal–Wallis test was performed to examine the abovementioned associations since the distributions were highly skewed; post hoc analysis was subsequently performed. In order to explore the dose-response relationships between the phthalate metabolites and microparticles, the extended model approach with multiple linear regression analyses was applied and potential confounders were adjusted: age, gender, hypertension, diabetes mellitus, total cholesterol, body mass index, smoking status, alcohol consumption, physical activity, and household income. The potential confounders consisted of both continuous variables (age, total cholesterol, BMI), and categorical variables (gender, hypertension, diabetes mellitus, smoking status, alcohol consumption, physical activity, and household income). Based on previous research, the abovementioned factors were associated with endothelial dysfunction or microparticles [32,33,34,35,36,37,38]; thus, all factors were selected in the regression model. A natural log transformation was performed for the concentrations of ΣDEHP, MnBP, EMPs, and PMPs. Further subgroup analyses were performed to examine whether the correlation between phthalate exposure and microparticle formation is affected by the abovementioned confounders. To explore the association between phthalate metabolites and microparticles, a Kruskal–Wallis test was performed to compare different levels of phthalate metabolites across four quartiles of microparticles, and also compare the highest to lowest quartiles. The trend test was modeled according to median value of each quartile of the phthalates to compare the difference of microparticles and was also adjusted for confounders; another set of same analyses were performed on non-diabetic participants. To examine the impact of co-exposure of the DEHP and DnBP on markers of endothelial dysfunction, these metabolites of ΣDEHP and MnBP were put in the same regression model. All statistical analyses were conducted using the SAS version 9.2 software by SAS Institute Inc. (Cary, NC, USA).

## 3. Results

Table 1 details the basic demographic characteristics of the study population and geometric means, including and 95% confidence interval (CI), of the concentrations of ΣDEHP and MnBP. The majority of our study cohort was female (n = 436, 62.6%), with an average age of 21.3 years (standard deviation, SD = 3.3). The mean concentration of creatinine-adjusted urinary ΣDEHP among all participants was 0.23 (95% CI = 0.21–0.25) μmol/g creatinine. The mean concentrations and 95% CI for MnBP was 38.99 (36.47–41.68) μg/g creatinine. Furthermore, the mean concentration of MnBP was statistically significantly higher among females (41.43 μg/g creatinine) compared to males (35.23 μg/g creatinine; *p* = 0.003). Diabetic participants had higher mean concentration of ΣDEHP compared to their counterparts (*p* = 0.038). Similar statistically significant differences were evident among those who are overweight (BMI ≥ 24), as this group of individuals had higher urinary concentrations of ΣDEHP and MnBP (*p* = 0.033 and 0.002, respectively).

Appendix A details the distribution and percentiles of the phthalates, and Appendix A outlines the geometric means and 95% CI of the concentrations of EMPs and PMPs of the study participants, according to basic characteristics. Those under 20 years of age had higher mean counts of EMPs and PMPs than those between the ages of 20 to 30 (with both *p*-values < 0.01). Males had a higher mean counts of EMPs than females (182.45 vs. 127.31 counts/μL, *p* < 0.01). Moreover, counts of EMPs were statistically significantly higher in those reported with hypertension, diabetes, hypercholesterolemia, and overweight (*p* ≤ 0.001, 0.03, 0.04, and <0.001, respectively).

In Table 2, the multivariate linear regression analysis showed that among males, per unit increase in the natural log-transformed concentration of MnBP was positively associated with higher natural log-transformed EMPs when compared to females. The subgroup analysis for different age groups showed that ΣDEHP levels were positively associated with counts of EMPs for those under 20 years of age and those between the ages of 20 to 30 (*p*-value 0.013 and <0.001; coefficient = 0.257 and 0.211, respectively). Similarly, MnBP levels were positively associated with counts of EMPs for those under 20 years of age and those between the ages of 20 to 30 (*p*-value 0.004 and <0.010; coefficient = 0.299 and 0.161, respectively).

We documented the geometric mean concentrations of creatinine-adjusted urinary ΣDEHP and MnBP, analyzed across increasing quartiles of serum counts of EMPs, in Table 3. Our results demonstrated that differences in levels of ΣDEHP and MnBP were statistically significant (*p* < 0.001), with urinary ΣDEHP and MnBP concentrations significantly increased with serum EMP counts (quartile 4 compared to quartile 1). The same analysis was also performed across quartiles of serum counts of PMPs, and results showed the concentrations of ΣDEHP significantly increased with serum PMP levels.

In Figure 2, the multivariable linear regression analysis showed that participants possessing higher quartiles of urinary ΣDEHP and MnBP presented with higher serum natural log-transformed EMPs than those in lower quartiles (*p* for trend <0.001 and <0.001; coefficient = 0.798 and 0.007; standard error = 0.189 and 0.001, respectively). Compared to the lowest quartile levels of MnBP, those in the highest quartile also had correspondingly higher counts of serum natural log-transformed EMPs, which appeared to have increased by more than 1.16-fold (*p*-value < 0.001). Furthermore, as concentrations of ΣDEHP increased across quartiles, the levels of natural log-transformed PMPs increased significantly (*p* for trend = 0.015); the abovementioned associations remain among non-diabetic participants (Appendix A). In Table 4, the multivariable linear regression analysis for co-exposures to DEHP and DnBP on both types of microparticles (EMPs and PMPs) showed that the increases in ΣDEHP and MnBP exposures from the 25th to the 75th percentile (i.e., an interquartile range change) were associated with the increases in EMPs by the multiplicative factors of 1.258 (95% CI: 1.122, 1.411) and 1.144 (95% CI: 1.022, 1.281), respectively. The increase in ΣDEHP was associated with the increase in PMPs by the multiplicative factor of 1.263 (95% CI: 1.072, 1.488), but MnBP was not associated with PMPs.

## 4. Discussion

Results yielded from our present study indicated significant positive associations existed between phthalate exposure and the formation of microparticles, which might be indicative of endothelial dysfunction. Specifically, our study showed as urinary concentrations of ΣDEHP and MnBP increased across quartiles, serum EMPs increased correspondingly. Moreover, increasing urinary concentrations of ΣDEHP was positively associated with higher serum PMPs levels. Since the formation of EMPs was symptomatic of cell apoptosis and the breakdown of cell-to-cell communication, and that of PMPs indicated stress and thrombus formation [39,40], both of which forewarn and can aggravate circulatory and cardiovascular diseases, our findings suggested exposure to DnBP might be associated with endothelial dysfunction, and such damage might manifest in the form of atherosclerotic-related vascular diseases in later life.

The present study focused on adolescents and young adults because this population may be potentially vulnerable to the toxic effects of phthalates. Since their organs are incompletely developed [41], the anatomical and physiological differences may affect the bioavailability of phthalates [42]. Furthermore, previous researchers indicated that exposure to DEHP and DnBP varied by age in adolescents and young adults [41,43,44]. For example, Fierens and colleagues indicated that exposure rate of DEHP increased with age, but the rate of DnBP decreased with age in Belgian population [43]. The dietary exposure to DEHP and DBP among different age group of children and adolescents in the Spanish population varied [41], and several consumer products were age group specific [42]. The aforementioned reported differences in dietary lifestyles or consumer products used in across age groups may explain these disparities on exposure. Furthermore, legislations about the use of phthalates may differ between countries, and these differences may also affect the exposure [43].

Previous in vivo animal studies reported that mice treated with DEHP through stimulating low-density lipoprotein oxidation presented with aggravated prognosis of hyperlipidemia, systemic inflammation, and atherosclerosis [45]. Our previous epidemiologic research reported that such link existed in young people, as we reported that DEHP and DnBP exposure was directly associated with increased risk of subclinical atherosclerosis, as well as coronary heart disease [27,46]. The results of our present study showed the aforementioned manifestations may be partially attributable to the increase in serum levels of EMPs and PMPs. Furthermore, evidence from our investigation imply that the increase in exposure to phthalates of both low and high molecular weight was positively correlated with increased concentrations of serum microparticle. Although such finding differs from previous literature [22], our study design echoes the recommendation of Kataria et al. and expanded the sample size. Furthermore, our study focused exclusively on phthalate exposure; therefore, we eliminated any potential interference brought on by bispenols. Although the exact biochemical mechanism remains uncertain, previous literature reported that MnBP may increase urinary biomarkers of oxidative stress, including 8-hydroxydeoxyguanosine and 8-isoprostane [47], which could explain the association between MnBP and endothelial dysfunction. Higher mean of urine MnBP concentrations was also reported to be associated with increased risk of stroke [48]. The possible reasons accounting for the association between MnBP and stroke include disturbance of the glycolytic pathway, which also distresses the production of cardiomyocytes and thus disrupts contractile function of the heart [49]. Although previous findings revealed an increased microparticle count to be associated with morbidities such as obesity, chronic kidney disease, and Henoch-Schonlein vasculitis, the mechanisms of microparticle formations remain elusive [50,51,52]. Additionally, prior reports indicated phthalates can induce cellular apoptosis [50,51]. Specifically, MEHP can adversely affect human umbilical vein endothelial cells via reactive oxygen species-mediated mitochondria-dependent pathway [53]. Since phthalates are known to induce oxidative stress, and oxidant stress contributes to endothelial dysfunction and arterial stiffness [22], we postulate that this may be the pathway through which exposure to plasticizers may contribute to endothelial dysfunction and the subclinical atherosclerotic in our young population. As previously reported in the research by Kataria and colleagues, the increase of high molecular weight phthalate metabolites was associated with altered levels of EMPs [22]. Trasande and colleagues also found that the increase of metabolites of DEHP was positively associated with albuminuria, which is an indicator of endothelial dysfunction [54]. Furthermore, it is worth noting that concentrations of DnBP metabolite found in our adolescents and young adults may be higher than that of other studies conducted considering that Taiwan banned the use of short-chain phthalate in children’s toys and care articles in 2011, which is much later compared to the USA, Canada, and the EU [55].

Since female and overweight participants presented with higher MnBP concentration, we postulate that a gender difference exists. This proposed notion was consistent with a previous report which found that both women (≥18 years) and girls (<18 years) had higher MnBP compared with males [44]. Although present study indicated that females had higher mean concentration of MnBP, the per unit increase in the concentration of MnBP among males was positively associated with higher level of EMPs when compared to females. Even though previous studies showed gender-related differences do not contribute to the number of circulating EMPs, both studies focused on middle-aged population, and not on adolescents and young adults [56,57]. Considering that gender may modify the associations between MnBP and cardiovascular risk factors [58], future research on how gender-specific differences affect microparticle formation and cardiotoxicity is warranted. Moreover, the present study found overweight subjects had higher MnBP and males were more vulnerable to DnBP exposure compared to females, and the findings were similar with Buser et al. [59] showing that childhood and adulthood with overweight had higher MnBP compared those with normal weight, and low molecular weight phthalate metabolites (e.g., MnBP) were associated with higher odds for obesity in male children and adolescents.

Limitations of this study should be noted. First, although results from our cross-sectional study revealed that meaningful associations existed between exposure to different types of phthalates and microparticle formation; a causal relationship should not be inferred. Second, findings of the present study pertained to adolescents and young adults and thus may not be generalized to other populations. Further studies are needed to examine the association and clinical implication between phthalate exposure and adverse cardiovascular outcome among the elderly and other susceptible groups. Third, although we adjusted for age, gender, BMI, and health behaviors (i.e., smoking and alcohol consumption), our results could still be confounded by pre-existing medical conditions, since a complete medical history on rheumatologic and hematologic disorders and malignancies were not examined in this study. However, this omission is unlikely to affect our majoring findings since more than 95% of the participants reported that they had no medical conditions or were not currently undergoing medical treatment at the time of the study. Fourth, other endocrine disrupting chemicals such as polychlorinated biphenyls or bisphenol A were not accounted for in this present study and their existence may influence the outcome of the present study. Finally, the present study included MEHP, MEHHP, MEOHP, but mono-(2-ethyl-5-carboxypentyl) phthalate and mono[2-(carboxymethyl)hexyl] phthalate were not included. Thus, the level of ΣDEHP in the present study might be underestimated.

## 5. Conclusions

We found the dose-response relationship exists between urinary concentrations of ΣDEHP and MnBP to both EMPs and PMPs, indicating that exposure to phthalates of both low and high-molecular weight might contribute to endothelial cell apoptosis, and such damage might induce inflammatory responses; thus, subjecting those exposed to higher risk of adverse cardiovascular outcome. Future in vivo and in vitro experiment studies are warranted to elucidate the relationship between phthalate exposure and EMPs and PMPs.

## Figures and Tables

**Figure 1 toxics-09-00033-f001:**
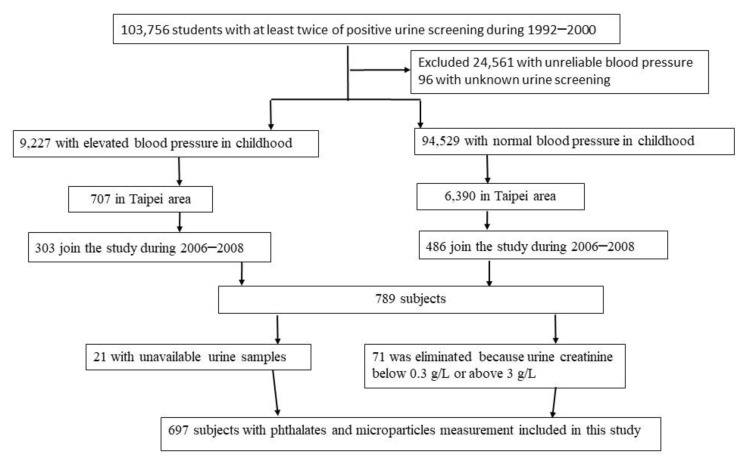
Flow chart of the study population recruitment.

**Figure 2 toxics-09-00033-f002:**
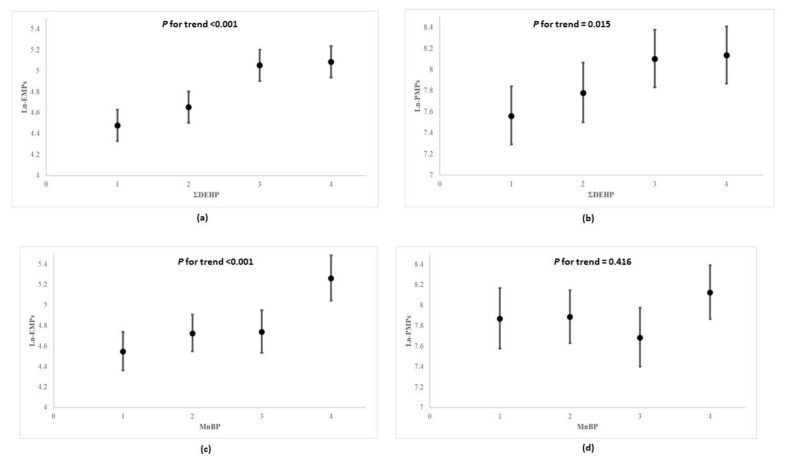
Geometric mean values and 95% confidence intervals of natural log-transformed EMPs and PMPs according to quartile concentrations of urinary ΣDEHP and MnBP ^a^ in multiple linear regression model ^b^. (**a**) ΣDEHP and Ln-EMPs, (**b**) ΣDEHP and Ln-PMPs, (**c**) MnBP and Ln-EMPs, (**d**) MnBP and Ln-PMPs. DEHP: di-2-ethylhexyl phthalate; MnBP: mono-n-butyl phthalate; EMP: endothelial microparticles; PMP: platelet microparticles. ^a^ ΣDEHP as μmol/g creatinine, and others as μg/g creatinine. EMPs and PMPs as counts/μL. ^b^ p for trend analysis—value was modeled according to median value of each quartile of each phthalates metabolites, and was adjusted for age, gender, body mass index, total cholesterol, hypertension, diabetes, smoking, alcohol consumption, physical activity, and household income.

**Table 1 toxics-09-00033-t001:** Basic characteristics of the sample subjects including geometric mean and 95% confidence interval of the creatinine-adjusted urinary ΣDEHP and MnBP ^a^.

Characteristics	N	ΣDEHP	MnBP
	697	0.23 (0.21–0.25)	38.99 (36.47–41.68)
Age			
12–19	204	0.25 (0.22–0.28)	40.01 (35.53–45.05)
20–30	493	0.22 (0.20–0.24)	38.58 (35.58–41.82)
Gender			
Male	261	0.20 ^c^ (0.17–0.22)	35.23 ^c^ (31.53–39.36)
Female	436	0.25 ^c^ (0.23–0.28)	41.43 ^c^ (38.12–45.03)
Hypertension			
Yes	62	0.27 (0.20–0.36)	43.84 (34.88–55.12)
No	635	0.23 (0.21–0.24)	38.55 (35.94–41.34)
Diabetes mellitus			
Yes	17	0.35 ^b^ (0.23–0.52)	45.78 (30.26–69.24)
No	680	0.23 ^b^ (0.21–0.25)	38.83 (36.29–41.56)
Total cholesterol (mg/dL)			
<200	154	0.21 (0.18–0.25)	41.96 (36.80–47.85)
≥200	543	0.23 (0.22–0.25)	38.19 (35.34–41.25)
Body mass index (kg/m^2^)			
<24	544	0.22 ^b^ (0.20–0.24)	36.64 ^c^ (33.94–39.55)
≥24	153	0.26 ^b^ (0.23–0.31)	48.65 ^c^ (42.67–55.47)
Smoking status			
Non-active smoker	579	0.24 (0.22–0.26)	39.84 (37.06–42.82)
Active smoker	118	0.21 (0.17–0.25)	35.09 (29.51–41.72)
Alcohol consumption			
Yes	44	0.21 (0.16–0.28)	39.79 (31.08–50.94)
No	653	0.23 (0.21–0.25)	38.94 (36.33–41.74)
Physical activity			
Yes	508	0.22 (0.20–0.24)	38.46 (35.57–41.59)
No	189	0.25 (0.22–0.29)	40.44 (35.54–46.03)
Household income			
<50,000 NTD/month	260	0.24 (0.21–0.27)	36.84 (32.54–41.71)
≥50,000 NTD/month	437	0.22 (0.21–0.25)	40.33 (37.34–43.55)

DEHP: di-2-ethylhexyl phthalate; MnBP: mono-n-butyl phthalate ^a^ Unit: ΣDEHP as μmol/g creatinine; MnBP = μg/g creatinine. ^b^
*p* < 0.05. ^c^
*p* < 0.01.

**Table 2 toxics-09-00033-t002:** Linear regression coefficients (standard error) of natural log-transformed endothelial microparticles (EMPs) and platelet microparticles (PMPs) with per unit increase in natural log-transformed ΣDEHP and MnBP in multiple linear regression model ^a^ for overall and subgroup population.

	Ln-ΣDEHP				Ln-MnBP			
	Ln-EMPs	*p*-Value	Ln-PMPs	*p*-Value	Ln-EMPs	*p*-Value	Ln-PMPs	*p*-Value
Overall	0.236 (0.046)	<0.001	0.186 (0.065)	0.005	0.208 (0.052)	<0.001	0.036 (0.074)	0.626
Age								
12–19	0.257 (0.102)	0.013	0.174 (0.129)	0.177	0.299 (0.103)	0.004	0.078 (0.131)	0.544
20–30	0.211 (0.053)	<0.001	0.171 (0.077)	0.026	0.161 (0.062)	0.010	−0.004 (0.089)	0.964
Gender								
Male	0.231 (0.073)	0.002	0.138 (0.110)	0.210	0.246 (0.082)	0.003	−0.045 (0.124)	0.714
Female	0.233 (0.060)	<0.001	0.220 (0.081)	0.007	0.183 (0.069)	0.008	0.083 (0.923)	0.367
Hypertension								
No	0.288 (0.049)	<0.001	0.171 (0.069)	0.014	0.210 (0.054)	<0.001	0.051 (0.077)	0.508
Yes	0.273 (0.166)	0.107	0.321 (0.214)	0.140	0.133 (0.226)	0.560	−0.063 (0.291)	0.829
Diabetes mellitus								
No	0.228 (0.046)	<0.001	0.177 (0.066)	0.007	0.208 (0.052)	<0.001	0.032 (0.074)	0.663
Yes	0.109 (1.052)	0.921	1.323 (1.397)	0.380	−0.324 (0.584)	0.599	0.177 (0.848)	0.842
Total cholesterol (mg/dL)								
<200	0.210 (0.051)	<0.001	0.141 (0.071)	0.049	0.188 (0.056)	<0.001	0.009 (0.078)	0.906
≥200	0.349 (0.114)	0.003	0.384 (0.167)	0.023	0.323 (0.140)	0.022	0.192 (0.205)	0.351
BMI status(kg/m^2^)								
<24	0.159 (0.053)	0.003	0.135 (0.072)	0.062	0.171 (0.058)	0.004	0.059 (0.080)	0.464
≥24	0.515 (0.103)	<0.001	0.432 (0.160)	0.008	0.406 (0.129)	0.002	0.008 (0.195)	0.967
Smoking status								
Non-active smoker	0.236 (0.052)	<0.001	0.158 (0.073)	0.032	0.180 (0.059)	0.002	0.0004 (0.082)	0.995
Active smoker	0.224 (0.107)	0.039	0.278 (0.159)	0.082	0.336 (0.116)	0.005	0.205 (0.177)	0.248
Alcoholic consumption								
No	0.228 (0.048)	<0.001	0.196 (0.066)	0.003	0.200 (0.053)	<0.001	0.049 (0.074)	0.509
Yes	0.360 (0.228)	0.123	−0.039 (0.404)	0.923	0.422 (0.256)	0.109	−0.225 (0.454)	0.623
Physical activity								
No	0.268 (0.082)	0.001	0.308 (0.115)	0.008	0.141 (0.097)	0.148	−0.010 (0.136)	0.940
Yes	0.222 (0.056)	<0.001	0.131 (0.079)	0.098	0.218 (0.063)	<0.001	0.030 (0.088)	0.730
Household income								
<50,000 NTD/month	0.242 (0.071)	<0.001	0.164 (0.101)	<0.105	0.167 (0.079)	<0.036	−0.082 (0.110)	0.459
≥50,000 NTD/month	0.232 (0.062)	<0.001	0.182 (0.087)	0.037	0.239 (0.071)	<0.001	0.102 (0.100)	0.308

DEHP: di-2-ethylhexyl phthalate; MnBP: mono-n-butyl phthalate; ^a^ Models were adjusted for age, gender, BMI, total cholesterol, hypertension, diabetes, smoking, alcohol consumption, physical activity, and household income.

**Table 3 toxics-09-00033-t003:** Geometric means and standard errors of creatinine-adjusted urinary concentration of ΣDEHP and MnBP by quartile distribution of EMPs and PMPs ^a^.

**Phthalate metabolites**	**EMPs**					
	Quartile 1 (n = 173)	Quartile 2 (n = 176)	Quartile 3 (n = 172)	Quartile 4 (n = 176)	*p*-1 value ^b^	*p*-2 value ^c^
	<52.86	52.86–145.71	145.71–364.29	>364.29		
ΣDEHP	0.206 (0.016)	0.194 (0.014)	0.226 (0.017)	0.310 (0.022)	<0.001	<0.001
MnBP	34.811 (2.425)	32.980 (2.291)	42.093 (2.723)	47.813 (3.096)	<0.001	<0.001
	**PMPs**					
	Quartile 1 (n = 173)	Quartile 2 (n = 176)	Quartile 3 (n = 172)	Quartile 4 (n = 176)	*p*-1 value ^b^	*p*-2 value ^c^
	<1062.86	1062.86–3628.57	3628.57–12040.0	>12040.0		
ΣDEHP	0.195 (0.015)	0.218 (0.015)	0.251 (0.020)	0.262 (0.020)	0.0150	0.0026
MnBP	36.250 (2.255)	40.35 (2.676)	41.861 (3.295)	37.704 (2.388)	0.4776	0.4826

DEHP: di-2-ethylhexyl phthalate; MnBP: mono-n-butyl phthalate; EMP: endothelial microparticles; PMP: platelet microparticles. ^a^ Unit: ∑DEHP as μmol/g creatinine, and others as μg/g creatinine. EMPs and PMPs as counts/μL. ^b^
*p*-1 value was Kruskal–Wallis test for medians. ^c^
*p*-2 value was for microparticles quartile 4 compared with quartile 1.

**Table 4 toxics-09-00033-t004:** Multiplicative factors (95% confidence intervals) for a change in serum endothelial and platelet microparticle level ^a^ associated with an interquartile range change in urinary phthalate metabolites level ^b^ in multiple linear regression model ^c^.

Phthalate Metabolites	Microparticles
	EMPs ^d^	PMPs ^d^
		*p*-value		*p*-value
ΣDEHP	1.258 (1.122, 1.411)	<0.001	1.263 (1.072, 1.488)	0.005
MnBP	1.144 (1.022, 1.281)	0.017	0.958 (0.816, 1.124)	0.594

## Data Availability

The data presented in this study are available on request from the corresponding author. The data are not publicly available due to privacy agreement signed by all participants.

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
