# Peer review of "Association between Urinary Phthalate Metabolites and Markers of Endothelial Dysfunction in Adolescents and Young Adults"

_toxics, 2021, doi:10.3390/toxics9020033_

Round 1

Reviewer 1 Report

This study was undertaken to study the associations between exposure to specific phthalates and circulating endothelial and platelet microparticles. There are many main weakness in this work, which are discussed below:

  1. In field 2.4, you only detected the MEHP, MEHHP, MEOHP. But the metabolites of DEHP also include the MECPP and the MCMHP. So the level of ∑DEHP you calculated might was lower than the real exposure level.
  2. This research only use the EMPs and PMPs as the markers of endothelial dysfunction and cardiovascular diseases was weak. You should add more outcome indicators.
  3. Only use the multivariate linear regression analysis to indicate the relationship between the DEHP metabolites and the EMPs and PMPs was unconvinced. You need do some in vivo and in vitro experiments to confirm this conclusion.
  4. The references were outdated and not enough to provide the latest support.

Author Response

Reviewer 1

This study was undertaken to study the associations between exposure to specific phthalates and circulating endothelial and platelet microparticles. There are many main weakness in this work, which are discussed below:

  1. In field 2.4, you only detected the MEHP, MEHHP, MEOHP. But the metabolites of DEHP also include the MECPP and the MCMHP. So the level of ∑DEHP you calculated might was lower than the real exposure level.

Response:

Thank you for the comments. The authors agree that the metabolites of DEHP include the MEHP, MEHHP, MEOHP, MECPP and the MCMHP. In the present study, we calculated the urinary level of these metabolites by summing the molar concentrations of MEHP, MEHHP, and MEOHP as previous research described [Ref. 1, Ref. 2]. We addressed the rationale of including MEHP, MEHHP, and MEOHP only, and the details are as follows: “To determine total exposure to DEHP, we estimated urinary concentrations of the metabolites by summing the molar concentrations of MEHP, MEHHP, and MEOHP as described in previous research (hereafter referred to as ΣDEHP in μmol/g creatinine) [28,29]”. Please see Method Section, line 145-147, page 4. We also listed this inclusion as a possible limitation of our study, and the details are as follows: “Finally, the present study included MEHP, MEHHP, MEOHP, but mono-(2-ethyl-5-carboxypentyl) phthalate and mono[2-(carboxymethyl)hexyl] phthalate were not included. Thus, the level of ∑DEHP in the present study might be underestimated.” (Discussion Section, line 343-346, page 11).

  1. This research only use the EMPs and PMPs as the markers of endothelial dysfunction and cardiovascular diseases was weak. You should add more outcome indicators.

Response:

We appreciate the suggestion. Besides the EMPs and PMPs, the authors agree that more outcome indicators of endothelial dysfunction and cardiovascular diseases are important to explore the association between phthalate exposure and endothelial dysfunction. Our present study decided to focus on EMPs and PMPs because they serve as early indicators of cardiotoxicity; we wanted to investigate whether exposure to certain phthalates may induce precursors of cardiotoxicity, therefore, we focused on EMPs and PMPs as the markers of endothelial dysfunction. For other outcome indicators, previous research published by our group examined the association between urinary MnBP concentrations and risk for coronary heart disease using high-sensitivity C-reactive protein, fibrinogen, and D-dimer [Ref. 3]. The results from this previous study indicated that significantly higher levels of the aforementioned indicators (high-sensitivity C-reactive protein, fibrinogen, and D-dimer) across the quartile of ΣDEHP in the group of coronary heart disease after adjustment for age, gender, body mass index, diabetes mellitus, hypertension, hypercholesterolemia, use of statins, smoking, and alcohol consumption. (p for trend=<0.001, 0.012, <0.001, respectively).

  1. Only use the multivariate linear regression analysis to indicate the relationship between the DEHP metabolites and the EMPs and PMPs was unconvinced. You need do some in vivo and in vitro experiments to confirm this conclusion.

Response:

Thank you for the comments. The authors agree that in vivo and in vitro studies are important methods to confirm the relationship between the DEHP metabolites and the EMPs and PMPs, and such types of studies can be carried out in future. The sentences related to the future work have been revised, and the details are as follows: (1) “Future in vivo and in vitro studies are warranted to elucidate whether a causal relationship exist between phthalate exposure and EMPs and PMPs.” Please see Abstact section, line 36-37, page 1;(2) “Future in vivo and in vitro experiment studies are warranted to elucidate the relationship between phthalate exposure and EMPs and PMPs.” Please see Conclusion section, line 352-353, page 11. The research design of the present study was a cross-sectional cohort, and the scope was to use the multivariate linear regression to examine whether a statistical significant association exist between phthalate exposure and microparticle formation. The sentences on the relationship have been revised, and the details are as follows: (1) “The findings indicate that exposures to phthalates of both low and high-molecular weight are positively associated with microparticle production, might contribute to endothelial dysfunction; such damage might manifest in the form of atherosclerotic-related vascular diseases.” Please see Abstract section, line 33-36, page 1;(2) “our findings suggested exposure to DnBP might be associated with endothelial dysfunction, and such damage might manifest in the form of atherosclerotic-related vascular diseases in later life.” Please see Discussion section, line 267-269, page 10; (3) “We found the dose-response relationship exist between urinary concentrations of ∑DEHP and MnBP to both EMPs and PMPs, indicating that exposure to phthalates of both low and high-molecular weight might contribute to endothelial cell apoptosis, and such damage might induce inflammatory responses; thus, subjecting those exposed to higher risk of adverse cardiovascular outcome.” Please see Conclusion section, line 348-352, page 11.

References of Response to Reviewer 1:

  1. Kim, Y.A.; Kho, Y.; Chun, K.C.; Koh, J.W.; Park, J.W.; Bunderson-Schelvan, M.; Cho, Y.H. Increased urinary phthalate levels in women with uterine leiomyoma: A case-control study. Int J Environ Res Public Health 2016, 13.
  2. Zhao, Y.; Shi, H.J.; Xie, C.M.; Chen, J.; Laue, H.; Zhang, Y.H. Prenatal phthalate exposure, infant growth, and global DNA methylation of human placenta. Environmental and molecular mutagenesis 2015, 56, 286-292.
  3. Su, T.C.; Hwang, J.J.; Sun, C.W.; Wang, S.L. Urinary phthalate metabolites, coronary heart disease, and atherothrombotic markers. Ecotoxicology and environmental safety 2019, 173, 37-44.

  1. The references were outdated and not enough to provide the latest support.

Response:

Thank you for comments. We have added five more references in 2018-2021, and the details are as follows:

“Based on previous research, the abovementioned factors were associated with endothelial dysfunction or microparticles [31-36]; thus, all factors were selected in the regression model.” Please see Method section, line 162-163, page 4. “Furthermore, previous researchers indicated that exposure to DEHP and DnBP varied by age in adolescents and young adults [39,41,42].” Please see Discussion section, line 273-274, page 10. “The dietary exposure to DEHP and DBP among different age group of children and adolescents in the Spanish population varied [39]”. Please see Discussion section, line 276-277,page 10. The references added are listed as follows (Please see Reference section, line 459-471 and 481-483, page 13-14):

  1. Yel, S.; Dursun, İ.; Çetin, F.; Baştuğ, F.; Tülpar, S.; Düşünsel, R.; Gündüz, Z.; Poyrazoğlu, H.; Yılmaz, K. Increased circulating endothelial microparticles in children with fmf. Biomarkers : biochemical indicators of exposure, response, and susceptibility to chemicals 2018, 23, 558-562.
  2. Pastore, I.; Bolla, A.M.; Montefusco, L.; Lunati, M.E.; Rossi, A.; Assi, E.; Zuccotti, G.V.; Fiorina, P. The impact of diabetes mellitus on cardiovascular risk onset in children and adolescents. International journal of molecular sciences 2020, 21.
  3. Oda, N.; Kajikawa, M.; Maruhashi, T.; Kishimoto, S.; Yusoff, F.M.; Goto, C.; Nakashima, A.; Tomiyama, H.; Takase, B.; Yamashina, A., et al. Endothelial function is preserved in light to moderate alcohol drinkers but is impaired in heavy drinkers in women: Flow-mediated dilation japan (fmd-j) study. PloS one 2020, 15, e0243216.
  4. Sangha, G.S.; Goergen, C.J.; Ranadive, S.M.; Prior, S.J.; Clyne, A.M. Preclinical techniques to investigate exercise training in vascular pathophysiology. American journal of physiology. Heart and circulatory physiology 2021.
  5. Lestido-Cardama, A.; Rodríguez Bernaldo de Quirós, A.; Bustos, J.; Lomo, M.L.; Paseiro Losada, P.; Sendón, R. Estimation of dietary exposure to contaminants transferred from the packaging in fatty dry foods based on cereals. Foods (Basel, Switzerland) 2020, 9.”.

Reviewer 2 Report

The proposed paper shows an investigation about phthalate exposure and endothelial dysfunction in a young population.

Overall the text is interesting, the method is well described. The results are presented with some completeness.

I have only few indications and suggestions to offer:

  • this is a cross-sectional study, this mean that you can produce only a hypothesis of causality as a result. The data to confirm this hypothesis can be produced only with case/control studies or prospective ones. This is not so clear in the text, starting from the abstract. You need to better clarify that your results lead to a POSSIBLE involvement of phthalate exposure in the entothelial dysfunction.
  • In Introduction you talk about the few literature articles on the theme, would be useful if you complete your considerations describing with more detailes these studies: sample characteristics? methods? what phthalates? results?
  • Your sample: is a numerous sample but really selected (in particular by age), and even if you specify that the results are on Young people I think is necessary to explain: why you chose this particular study population? are young people more exposed to phthalates (I don't think so)? Are Young people more involved in endothelial dysfunction? Please clarify the choice and, in discussion, I think is needed to shedding light to the opportunity of the involment in future Studies of a population with a higher age range (also with consideration about the higher incidence of cardiovascular disease in older people)
  • Results: the linear regression result could be better illustrated with a graph instead than table 4
  • Results: line 158-160 you indicate that, for MnBP, there's a significat difference between male and female, but is not given the p value. (in methods you  need to clarify what test was used to evaluate the statistic significant difference of mean values)
  • Finally: some typo are present in the text, to be corrected: i.e line 64 (OR instead than OF) or line 20 ( Epidemiologic instead than epidemiologic). Proofread the Whole text.

Author Response

Reviewer 2

The proposed paper shows an investigation about phthalate exposure and endothelial dysfunction in a young population.

Overall the text is interesting, the method is well described. The results are presented with some completeness.

I have only few indications and suggestions to offer:

  • this is a cross-sectional study, this mean that you can produce only a hypothesis of causality as a result. The data to confirm this hypothesis can be produced only with case/control studies or prospective ones. This is not so clear in the text, starting from the abstract. You need to better clarify that your results lead to a POSSIBLE involvement of phthalate exposure in the entothelial dysfunction.

Response:

Thank you for the comments. The authors agree that a cross-sectional study design can be used to demonstrate correlation between exposure and outcome, and such research design cannot confirm causality. The authors agree that the present results only lead to a possible involvement of phthalate exposure in the endothelial dysfunction. We have revised the sentences, and the details are as follows: (1) “In conclusion, our results demonstrate a potential dose-response relationship between exposures to phthalates (∑DEHP and MnBP) and microparticle formation (EMPs and PMPs) in adolescents and young adults. The findings indicate that exposures to phthalates of both low and high-molecular weight are positively associated with microparticle production, might contribute to endothelial dysfunction; such damage might manifest in the form of atherosclerotic-related vascular diseases.” Please see Abstract section, line 31-36, page 1. (2) “Results yielded from our present study indicates significant positive association exist between phthalate exposure and the formation of microparticles, which might be indicative of endothelial dysfunction.” Please see Discussions section, line 260-262, page 10. (3) “our findings suggested exposure to DnBP might be associated with endothelial dysfunction, and such damage might manifest in the form of atherosclerotic-related vascular diseases in later life.” Please see Discussions section, line 267-269, page10. (4) “We found the dose-response relationship exist between urinary concentrations of ∑DEHP and MnBP to both EMPs and PMPs, indicating that exposure to phthalates of both low and high-molecular weight might contribute to endothelial cell apoptosis, and such damage might induce inflammatory responses; thus, subjecting those exposed to higher risk of adverse cardiovascular outcome.” Please see Conclusions section, line 348-352, page 11.

  • In Introduction you talk about the few literature articles on the theme, would be useful if you complete your considerations describing with more detailes these studies: sample characteristics? methods? what phthalates? results?

Response:

We appreciate the suggestion. Details regarding previous exposure studies such as sample characteristics, types of phthalates exposed, and results have been included in the revised introduction, and the details are as follows; “Chen et al. reported developmental toxicity from DBP and estrogenic endocrine disrupting activity from DEHP and DBP in zebrafish embryos. Furthermore, these phthalates could cause additional toxicity symptoms such as cardio edema [5]. Another animal toxicity study revealed that rats chronically exposed to high oral doses of DEHP exhibited many noncarcinogenic effects such as changes in clinical hematology, and manifestations of increased liver lesions [7]. DEHP exposure during lactational period was also noted to impair insulin signal transduction and oxidation in the cardiac muscle of female progeny rats [8]. Moreover, in utero exposure to DEHP caused long-term cardiovascular effects in male offspring rats [9]. In additional to animal experiments, systemic review of both epidemiological and animal toxicity studies concluded that DEHP and DBP exposure may lead to adverse reproductive health outcomes in females [6]. Additionally, the biochemical mechanism through which cardiotoxicity occurs have been reported. DEHP exposure in mice can induce oxidative stress, inhibit acetylcholinesterase activity, and lead to structural alternation of surrounding blood vessels, as reported in previous literature [10]; these disruptions provide evidence that exposure to DEHP may promote the progression of atherosclerosis. Furthermore, Ferguson and associates indicated that mono-n-butyl phthalate (MnBP), the metabolite of di-n-butyl phthalate (DnBP) and the metabolites of DEHP, was significantly and positively associated with markers of oxidative stress (bilirubin) and inflammation (fibrinogen, alkaline phosphatase) based on data from participants in the National Health and Nutrition Examination Survey [11].”. Please see Introduction section, line 52-70, page 2.

  • Your sample: is a numerous sample but really selected (in particular by age), and even if you specify that the results are on Young people I think is necessary to explain: why you chose this particular study population? are young people more exposed to phthalates (I don't think so)? Are Young people more involved in endothelial dysfunction? Please clarify the choice and, in discussion, I think is needed to shedding light to the opportunity of the involment in future Studies of a population with a higher age range (also with consideration about the higher incidence of cardiovascular disease in older people)

Response:

We appreciate the suggestion. We have added the choice of study population in Discussion section, and the details are as follows: “The present study focused on adolescents and young adults because this population may be potentially vulnerable to the toxic effects of phthalates. Since their organs are incompletely developed [39], the anatomical and physiological differences may affect the bioavailability of phthalates [40]. Furthermore, previous researchers indicated that exposure to DEHP and DnBP varied by age in adolescents and young adults [39,41,42]. For example, Fierens and colleagues indicated that exposure rate of DEHP increased with age, but the rate of DnBP decreased with age in Belgian population [41]. The dietary exposure to DEHP and DBP among different age group of children and adolescents in the Spanish population varied [39], and several consumer products are age group specific [40]. The aforementioned reported differences in dietary lifestyles or consumer products used in across age groups may explain these disparities on exposure. Furthermore, legislations about the use of phthalates may differ between countries, and these differences may also affect the exposure [41].” Please see Discussion section, line 270-280, page 10. About the involvement of young people in endothelial dysfunction, we also added the following statements to the discussion section: “As previously reported in the research by Kataria and colleagues, increase of high molecular weight phthalate metabolites was associated with altered levels of EMPs [20]. Trasande and colleagues also found that the increase of metabolites of DEHP was positively associated with albuminuria, which is an indicator of endothelial dysfunction [54].” Please see Discussion section, line 307-311, page 10.

  • Results: the linear regression result could be better illustrated with a graph instead than table 4

Response:

We appreciate the suggestion. We have changed table 4 into a graph (Figure 2) as suggested. Please see the new Figure 2, line 241-251, page 9.

  • Results: line 158-160 you indicate that, for MnBP, there's a significat difference between male and female, but is not given the p value. (in methods you need to clarify what test was used to evaluate the statistic significant difference of mean values)

Response:

We appreciate the suggestion. The p-value has been added, and the details are as follows: “Furthermore, the mean concentration of MnBP was statistically significantly higher among females (41.43 μg/g creatinine) compared to males (35.23 μg/g creatinine; p=0.003).”. Please see Results section, line 181-183, page 5. About the statistical methods, we have revised the tests used to evaluate the statistic significant difference. The details are as follows: “In order to explore the dose-response relationships between the phthalate metabolites and microparticles, the extended model approach with multiple linear regression analyses was applied and potential confounders were adjusted age, gender, hypertension, diabetes mellitus, total cholesterol, body mass index, smoking status, alcohol consumption, physical activity, and household income. The potential confounders consisted of both continuous variables (age, total cholesterol, BMI), and categorical variables (gender, hypertension, diabetes mellitus, smoking status, alcohol consumption, physical activity, and household income). Based on previous research, the abovementioned factors were associated with endothelial dysfunction or microparticles [31-36]; thus, all factors were selected in the regression model. A natural log transformation was performed for the concentrations of ΣDEHP, MnBP, EMPs, and PMPs. Further subgroup analyses were performed to examine whether the correlation between phthalate exposure and microparticle formation is affected by the abovementioned confounders. To explore the association between phthalate metabolites and microparticles, Kruskal-Wallis test was performed to compare different levels of phthalate metabolites across four quartiles of microparticles, and also compare the highest to lowest quartiles. The trend test was modeled according to median value of each quartile of the phthalates to compare the difference of microparticles and was also adjusted for confounders; another set of same analyses were performed on non-diabetic participants. To examine the impact of co-exposure of the DEHP and DnBP on markers of endothelial dysfunction, these metabolites of ΣDEHP and MnBP were put in the same regression model.” Please see Method section, line 155-173, page 4-5.

  • Finally: some typo are present in the text, to be corrected: i.e line 64 (OR instead than OF) or line 20 ( Epidemiologic instead than epidemiologic). Proofread the Whole text.

Response:

We appreciate the suggestion. The typos on line 20 and 64 have been corrected, and please see line 20 and line 75. Additionally, the entire manuscript have been proofread.

Reviewer 3 Report

This study examined the associations between exposure to specific phthalates and circulating endothelial and platelet microparticles in adolescents and young adults. Although this manuscript is of interest, I have several comments as follows:

  1. The authors could describe which one was the continuous or categorical variable in these potential confounders in the statistical section and how to select these covariates in the models.
  2. The authors did not present the results of total cholesterol and diabetes in table 3. Were all variables included in the same model in table 3? The authors could consider describing these for more detail.
  3. The concentrations of ∑DEHP and MnBP did not significantly increase with serum EMPS and PMPS in table 3 according to p for trend test. Furthermore, these results for p for trend in table 3 were conflicting as compared to those in table 4. The authors could check it carefully.
  4. The authors described that this beta coefficients (such as 329.472) for p for trend test was too big in table 4. Were phthalate metabolites levels and/ or outcomes transformed in these models?
  5. The authors could consider putting these metabolites levels (such as ∑DEHP ad MnBP) in the same model to examine the impact of co-exposure on markers of endothelial dysfunction.
  6. Because serum EMPS and PMPS in adolescents were significantly higher than those in young adults, the authors could consider performing these analyses stratified by age group.

Author Response

Reviewer 3

This study examined the associations between exposure to specific phthalates and circulating endothelial and platelet microparticles in adolescents and young adults. Although this manuscript is of interest, I have several comments as follows:

  1. The authors could describe which one was the continuous or categorical variable in these potential confounders in the statistical section and how to select these covariates in the models.

Responses:

We appreciate the suggestion. The characteristics of variables have been added in the statistical section, and the details are as follows: (1) “The characteristics included the following: age, gender, hypertension, diabetes mellitus, total cholesterol, body mass index, smoking status, alcohol consumption, physical activity, and household income.” Please see Method section, line 150-152, page 4. (2) “The potential confounders consisted of both continuous variables (age, total cholesterol, BMI), and categorical variables (gender, hypertension, diabetes mellitus, smoking status, alcohol consumption, physical activity, and household income).” Please see Method section, line 159-161, page 4. The aforementioned covariates were selected because previous research indicated they were associated with endothelial dysfunction or microparticles; thus, all factors were selected in the regression model. We have added the reason in the statistical section, and the details are as follows: “Based on previous research, the abovementioned factors were associated with endothelial dysfunction or microparticles [31-36]; thus, all factors were selected in the regression model.” Please see Method section, line 162-163, page 4.

  1. The authors did not present the results of total cholesterol and diabetes in table 3. Were all variables included in the same model in table 3? The authors could consider describing these for more detail.

Responses:

We appreciate the suggestion. About the presentation of the results of total cholesterol and diabetes, because Table 3 is not the model analysis, but Table 2 is the model analysis. Thus, I think you mean Table 2, instead of Table 3. The results of total cholesterol and diabetes mellitus have been added in Table 2. All variables, including age, gender, BMI, total cholesterol, hypertension, diabetes, smoking, alcohol consumption, physical activity, and household income, have included in the same model in Table 2. Please see the revised Table 2 in Result section, line 208-213, page 7-8.

  1. The concentrations of ∑DEHP and MnBP did not significantly increase with serum EMPSand PMPin table 3 according to p for trend test. Furthermore, these results for p for trend in table 3 were conflicting as compared to those in table 4. The authors could check it carefully.

Responses:

We appreciate the reminder. The statistical method for the p for trend test in Table 3 was to use the ∑DEHP and MnBP as the outcomes, and we think it is inappropriate based on the hypothesis that phthalate exposure might induce increase of the EMPs and PMPs. In contrast, the statistical method for the p for trend test in Table 4 was to use the EMPs and PMPs as the outcomes, and it is appropriate. Thus, we have deleted the p for trend test in Table 3. We remained the results of p-1 value in Table 3, which compared the medians of ∑DEHP and MnBP across four quartiles of the EMPs and PMPs. We also remained the results of p-2 value in Table 3, which compared the medians of ∑DEHP and MnBP between quartiles 1 and quartiles 4 of the EMPs and PMPs. Please see the revised Table 3 in Result section, line 221-226, page 7.

In addition, multiple linear regression model for overall population, which was to examine natural log-transformed EMPs and PMPs with per unit increase in natural log-transformed ∑DEHP and MnBP, have been added to Table 2, and the results showed that the concentrations of ∑DEHP and MnBP were positively associated with serum EMPs level, and the concentration of ∑DEHP was positively associated with serum PMPs level. The abovementioned associations were consistent with Table 4. Please see the revised Table 2 in Result section, line 208-213, page 7-8.

  1. The authors described that this beta coefficients (such as 329.472) for p for trend test was too big in table 4. Were phthalate metabolites levels and/ or outcomes transformed in these models?

Responses:

We appreciate the suggestion. In Table 4, one possible reason accounting for the big beta coefficients for the association between urinary concentrations of ∑DEHP and serum EMPs level is that the range of EMPs level was wide, and the maximal and median values was 5715.714, and 145.714, respectively; the median level of ∑DEHP was relative small, which was 0.209. Thus, the EMPs and PMPs have been natural log-transformed, and the new results showed in the new Figure 2 (To be better illustrated with graph, one reviewer suggests to change table 4 into a graph). Please see the new Figure 2 in Result section, line 241-251, page 9. The revised results were similar, and the details are as follows: “In Figure 2, the multivariable linear regression analysis showed that participants possessing higher quartiles of urinary ∑DEHP and MnBP present with higher serum natural log-transformed EMPs than those in lower quartiles (p for trend <0.001 and <0.001; coefficient=0.798 and 0.007; standard error = 0.189 and 0.001, respectively). Compared to the lowest quartile levels of MnBP, those in the highest quartile also have correspondingly higher counts of serum natural log-transformed EMPs, which appear to have increased by more than 1.16-fold (p-value< 0.001). Furthermore, as concentrations of ∑DEHP increases across quartiles, the levels of natural log-transformed PMPs increases significantly (p for trend=0.015);”. Please see Result section, line 227-234, page 8. The similar sentence in the abstract have also revised, and the details are as follows: “Multiple linear regression indicated that increases in urinary concentrations of ∑DEHP and MnBP (mono-n-butyl phthalate), across quartiles, were positively associated with serum EMPs level (P for trend <0.001 and <0.001; β=0.798 and 0.007; standard error=0.189 and 0.001, respectively).” Please see Abstract section, line 26-29, page 1. In addition, in supplementary Table 3 for non-diabetic participants, the EMPs and PMPs have also been natural log-transformed, and the supplementary Table 3 has been revised. The results were similar, and please see the revised supplementary Table 3 in supplementary file.

  1. The authors could consider putting these metabolites levels (such as ∑DEHP ad MnBP) in the same model to examine the impact of co-exposure on markers of endothelial dysfunction.

Responses:

We appreciate the suggestion. We added the multivariable linear regression analysis for co-exposure of the ∑DEHP and DnBP on micropartivles (EMPs and PMPs), and please see the new Table 5 in Result section, line 252-258, page 9. The results showed that the levels of urinary ∑DEHP and MnBP were positively associated with levels of serum EMPs (p-value <0.001 and 0.017; coefficient=0.196 and 0.131, respectively). The levels of urinary ∑DEHP was also positively associated with levels of serum PMPs (p-value 0.005; coefficient=0.199), but the MnBP was not associated with the PMPs. The details of the revised statistical analysis are as follows: “To examine the impact of co-exposure of the DEHP and DnBP on markers of endothelial dysfunction, these metabolites of ΣDEHP and MnBP were put in the same regression model.” Please see Method section, line 171-173, page 5. The details of the revised results are as follows:“In Table 5, the multivariable linear regression analysis for co-exposures to DEHP and DnBP on both types of microparticles (EMPs and PMPs) showed that that levels of urinary ∑DEHP and MnBP were positively associated with levels of serum EMPs (p-value <0.001 and 0.017; coefficient=0.196 and 0.131, respectively). The levels of urinary ∑DEHP was positively associated with levels of serum PMPs (p-value 0.005; coefficient=0.199), but concentrations of MnBP was not associated with counts of PMPs.”. Please see Results section, line 235-240, page 8-9.

  1. Because serum EMPSand PMPin adolescents were significantly higher than those in young adults, the authors could consider performing these analyses stratified by age group.

Responses:

We appreciate the reminder. We performed subgroup analysis stratified by age and presented the results in Table 2. The sentences of the results have been added, and the details are as follows: “The subgroup analysis for different age groups showed that ∑DEHP levels were positively associated with counts of EMPs for those under 20 years of age and those between the ages of 20 to 30 (p-value 0.013 and <0.001; coefficient=0.257 and 0.211, respectively). Similarly, MnBP levels were positively associated with counts of EMPs for those under 20 years of age and those between the ages of 20 to 30 (p-value 0.004 and <0.010; coefficient=0.299 and 0.161, respectively).”. Please see Result section, line 201-206, page 6.; Table 2, line 208-213, page 7-8.

Round 2

Reviewer 1 Report

Thanks for authors' response. But I still think the weakness I mentioned has not been corrected.

  1. Although the authors have disscussed that “the present study included MEHP, MEHHP, MEOHP, but mono-(2-ethyl-5-carboxypentyl) phthalate and mono[2-(carboxymethyl)hexyl] phthalate were not included. Thus, the level of ∑DEHP in the present study might be underestimated”, the weakness still exists. If included the MECPP and the MCMHP, the results would be changed. So I think it's quite necessary to include all metabolites.
  2.  Authors have mentioned that the in vivo and the in vitro studies can be carried out in future, but without these studies' results, this research was incomplete. Moreover, without in vivo and the in vitro studies, this research was too weak to clarify the relationship between the metabolites and the biomarkers. 

Author Response

1.Although the authors have disscussed that “the present study included MEHP, MEHHP, MEOHP, but mono-(2-ethyl-5-carboxypentyl) phthalate and mono[2-(carboxymethyl)hexyl] phthalate were not included. Thus, the level of ∑DEHP in the present study might be underestimated”, the weakness still exists. If included the MECPP and the MCMHP, the results would be changed. So I think it's quite necessary to include all metabolites.

Response:

Thank you for the comments. The authors agree that the weakness of not including the MECPP and the MCMHP exists, and we have mentioned this issue as a possible limitation of our study.

2. Authors have mentioned that the in vivo and the in vitro studies can be carried out in future, but without these studies' results, this research was incomplete. Moreover, without in vivo and the in vitro studies, this research was too weak to clarify the relationship between the metabolites and the biomarkers.

Response:

Thank you for the comments. The authors agree that in vivo and in vitro studies are important methods to confirm the relationship between the DEHP metabolites and the EMPs and PMPs, and such types of studies can be carried out in future. The research design of the present study was a cross-sectional human cohort, and the scope of this study is not feasible to do in vitro or in vivo experimental work. The present results may be helpful for further in vitro or in vivo experimental studies.